# CO-EVOLUTION LEARNING

## ABSTRACT

Generative and representation models, whether trained independently or evolved separately, require high-quality, diverse training data, imposing limitations on their advancement. Specifically, self-supervised learning, as a popular paradigm for representation learning, decreases the reliance on labeled data in representation models. However, it still necessitates large datasets, specialized data augmentation techniques, and tailored training strategies. While generative models have shown promise in generating diverse data, ensuring semantic consistency is still a challenge. This paper introduces a novel co-evolution framework (referred to as CORE) designed to address these challenges through the mutual enhancement of generative and representation models. Without incurring additional, unacceptable training overhead compared to independent training, the generative model utilizes semantic information from the representation model to enhance the quality and semantic consistency of generated data. Simultaneously, the representation model gains from the diverse data produced by the generative model, leading to richer and more generalized representations. By iteratively applying this co-evolution framework, both models can be continuously enhanced. Experiments demonstrate the effectiveness of the co-evolution framework across datasets of varying scales and resolutions. For example, implementing our framework in LDM can reduce the `FID` from 43.40 to 20.13 in unconditional generation tasks over the ImageNet-1K dataset. In more challenging scenarios, such as tasks with limited data, this framework significantly outperforms independent training of generative or representation model. Furthermore, employing the framework in a self-consuming loop effectively mitigates model collapse. Our code will be publicly released.

> **Quotation 1 .** *"What I cannot create, I do not understand."* — Richard Feynman.

## 1 INTRODUCTION

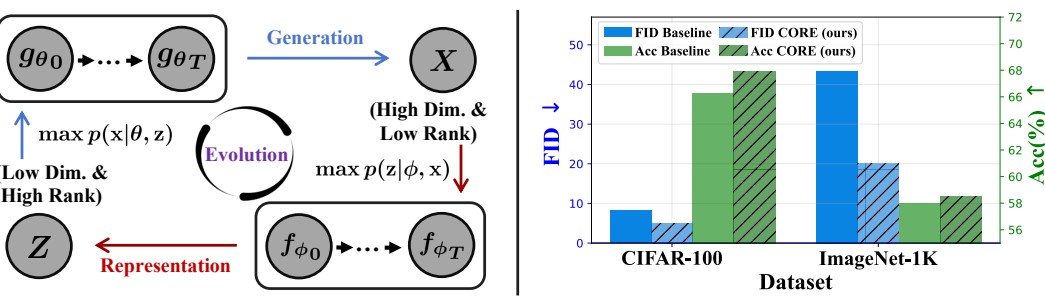

Figure 1: (a) The left panel illustrates the process and key components of our proposed framework. Generated samples and features ($D_X$ and $D_Z$) assist in training representation and generative models ($g_\theta$ and $f_\phi$). These trained models subsequently generate higher-quality features and samples to assist each other, facilitating iterative co-evolution within the framework. (b) The right panel presents the experimental results for the baseline models and those enhanced with our CORE, following one evolution round. The generative model employed is ADM, and the representation model is SimCLR, utilizing ResNet-50 as the backbone. These results demonstrate the efficacy of our CORE in boosting performance and improving training efficiency for baseline models.

Deep representation models (Bai et al., 2023; Achiam et al., 2023; Kirillov et al., 2023), designed to extract semantic information from real-world data, have made substantial advances across a range of

downstream applications (Han et al., 2024; Zhou et al., 2023; Wang et al., 2024a; Liu et al., 2024a; Ma et al., 2024; Zou et al., 2024; Lai et al., 2024; Liu et al., 2024b; Yin et al., 2023; Deitke et al., 2024; Yang et al., 2024; Mazurowski et al., 2023; Brohan et al., 2023; Hong et al., 2023; Ke et al., 2024; Wei et al., 2023; Chen et al., 2024a; Huang et al., 2024). Simultaneously, generative models (Yang et al., 2023; Ho et al., 2020; Song et al., 2020; Nichol & Dhariwal, 2021; Choi et al., 2021; Li et al., 2024; Chen et al., 2024b; Li et al., 2023), which prioritize the generation of high-quality, realistic data, have also achieved notable success within the deep learning community.

However, both generative models and representation models yet face several real-world challenges. In the *challenge (i)* of generative models: the SoTA generative models such as diffusion models (Song et al., 2020; Ho et al., 2020) requires training on highly diverse and high-quality data (Rombach et al., 2022), where a lack of diversity or quality in the data can prevent the models from accurately capturing the underlying distribution, resulting in issues such as mode collapse (Shumailov et al., 2024). Such extensive training datasets together with the large neural model architectures involved (Yang et al., 2023) therefore require significant computational resources, yet the trained model often generates samples with incorrect semantic information, contradicting real-world physical observations. Similarly, in the *challenge (ii)* of representation models, a diverse and high-quality labeled data is the key. Though self-supervised learning (Chen et al., 2020; Oquab et al., 2023; Chen et al., 2021) reduces the need for labeled datasets, it requires significantly larger and diverse datasets along with carefully designed data augmentation and training strategies (Chen et al., 2020), encountering significant computational costs.

Building on previous studies (Huh et al., 2024; Sun et al., 2024) that as model sizes expand and training tasks diversify, distinct models tend to converge on similar representations, we observe a consistent trend toward a shared statistical model capable of capturing the fundamental structure of real-world phenomena. We further extend this insight and argue that *deep representation and generative models essentially perform the same task*: capturing the underlying distribution and structural attributes of real-world data, whether for generation or representation.

To this end, we draw inspiration from the double helical structure of DNA and propose a helical co-evolution strategy for generation and representation. This approach simultaneously addresses *challenge (i)* and *challenge (ii)*: it leverages representation models to enhance generative ability models to capture the distribution of real-world data, and employs generative models to produce high-quality yet diverse data to improve the training of representation models. Specifically, we propose three technical frameworks: (1) **R2G Framework** leverages a representation model to generate semantic features $\mathbf{z}$ for each input sample $\mathbf{x}$. These features are used as additional guidance to train a generative model by maximizing $p(\mathbf{x}|\boldsymbol{\theta}, \mathbf{z})$. (2) **G2R Framework** employs a generative model to generate diverse samples $\mathbf{x}'$ from the original input $\mathbf{x}$. The generated samples are utilized to improve the representation model by optimizing $p(\mathbf{z}'|\boldsymbol{\phi}, \mathbf{x}')$. (3) **CORE Framework** built on frameworks (1) and (2), the co-evolution framework iteratively strengthens both the generative and representation models through a feedback loop, progressively enhancing their performance. We present the following **five key contributions below**, laying the groundwork for a co-evolution paradigm in representation and generative learning.

(a) Intriguingly, we find that even a lightly trained representation model effectively triggers our R2G framework. To illustrate, employing a representation model trained on CIFAR-100 (Krizhevsky et al., 2009) requiring less than 16 GPU-minutes, we can significantly improve the training of ADM (Nichol & Dhariwal, 2021), leading to a $30\%$ reduction in the `FID` score. For further details, refer to Section 4.1 .

(b) By employing a pre-trained generative model from R2G, we can generate diverse and realistic samples that improve the training and recognition performance of existing representation models.

(c) By identifying the efficacy of mutual assistance between representation and generative models, we introduce the task of co-evolution between generative and representation models and propose a straightforward framework (see CORE in Section 3 ) to facilitate this. Notably, there is minimal additional training overhead in each iteration of the co-evolution training loop.

(d) Our extensive experiments indicate that applying (a) R2G, (b) G2R, or (c) CORE within the training of generative or representation models effectively enhances their performance. These experiments are conducted across various datasets and tasks (see Section 4.2 and Section 4.3 ).

(e) Additionally, we apply this co-evolution task in both data-scarce and self-consuming loop scenarios, demonstrating the significant potential of this method to address real-world challenges and provide novel insights to the community (see Section 4.5 and Section 4.6 ).

## 2 RELATED WORK

Existing research has extensively investigated the unidirectional support between representation models and generative models. Our co-evolutionary framework, however, introduces two fundamental distinctions: (a) It functions as a multi-round, iterative process rather than a single-round one. (b) It alleviates the pressure on individual model design, enabling the inclusion of suboptimal models and incorporating the training process into the co-evolutionary framework.

In contrast, prior uni-directional approaches often necessitate a robust, well-trained model as a basis (Li et al., 2023; Wang et al., 2024b), typically requiring significant data and computational resources that might be difficult to access.

Our framework is constructed on advanced generative and representation models. Specifically, we emphasize diffusion models for generative tasks (Sohl-Dickstein et al., 2015; Ho et al., 2020; Song & Ermon, 2019), which are now the dominant paradigm in image generation. Below, we examine how representation models support generative models and vice versa.

**Generative models.** Generative models have become pivotal in artificial intelligence, enabling the creation of synthetic data with impressive realism and diversity. Deep learning has notably advanced this field through the introduction of Generative Adversarial Networks (GANs) (Goodfellow et al., 2014) and Variational Autoencoders (VAEs) (Kingma, 2013), significantly enhancing image generation capabilities. However, GANs encounter several architectural challenges (Dhariwal & Nichol, 2021), primarily arising from the unstable nature of training the generator and discriminator simultaneously. This instability often leads to mode collapse, where the generator produces uniform samples. Diffusion models, inspired by non-equilibrium thermodynamics, present an alternative by increasing system entropy over time to enhance randomness. Recent advancements by OpenAI have substantially improved the practicality of diffusion models in real-world applications (Song et al., 2023; Yang et al., 2023).

**Generative model training/sampling with representation models.** Using an external representation model to assist a generative model enables more controllable generative processes (Ramesh et al., 2022; Nichol et al., 2022; Rombach et al., 2022). By leveraging a pre-trained representation model, control information can be extracted and used as a condition during the training of the generative model. Additionally, gradient information from the discriminative model can guide category-related sampling, further enhancing control over the generation process (Dhariwal & Nichol, 2021).

To obtain a representation to assist in training a conditioned generative model, a self-supervised model like SimCLR (Chen et al., 2020) and DINO (Caron et al., 2021), or supervised model like CLIP (Radford et al., 2021) can be directly used, where the model outputs an embedding vector that serves as the conditioning information. These representation vectors can be further clustered to obtain cluster-level information (Adaloglou et al., 2024; Hu et al., 2023), or multiple neighboring representations can be leveraged for enhanced control (Blattmann et al., 2022). Regarding the granularity of the conditional information obtained, finer-grained conditional signals are also possible beyond image-level representations, such as bounding boxes and segmentation maps (Hu et al., 2023; Luo et al., 2024).

Recently, the Representation Diffusion Model (RDM) proposed by RCG (Li et al., 2023) has been introduced to model the representation space effectively. In the training of generative models, particularly pixel generators, the output from a well-trained self-supervised encoder is utilized as the conditioning input. This approach by RCG shows superior performance in non-label-conditioned generation, improving training on unlabeled datasets. However, the complexity of the framework requires additional training for the RDM. In contrast, our proposed R2G method eliminates the need for a well-trained representation model and avoids the additional training required for the RDM.

**Training representation models with generated data.** Generative models can produce virtually limitless samples for downstream model training. Leveraging advanced Diffusion Models to assist in the training of representation models has led to significant performance gains across various tasks (Wang et al., 2024b; Tian et al., 2024a;b; Jahanian et al., 2021; Afkanpour et al., 2024). Existing studies have demonstrated the potential of synthesizing data through generative models in supervised learning (Azizi et al., 2023; He et al., 2022; Sarıyıldız et al., 2023), self-supervised learning (Chen et al., 2024b), and adversarial training (Wang et al., 2023). By using label information from reference datasets to generate image data, or by combining large language models (LLMs) with image generation models to produce vast amounts of image-text paired data (Hammoud et al.,

2024), it is possible to achieve results that match or even surpass those of models trained on real data. AdaInf (Wang et al., 2024b) exemplifies how generative models can enhance model characterization. It utilizes 1 million images generated by a generative model to augment the original training data. Through data inflation, reweighting, and weak augmentation techniques, AdaInf achieves improved self-supervised learning performance by effectively blending synthetic and original data.

To generate diverse and realistic samples for training, AdaInf requires a well-trained and powerful generative model. However, accessing such a model is typically challenging, especially for complex datasets like ImageNet-1K (Deng et al., 2009).

## 3 METHODOLOGY

Our research introduces a novel approach to facilitate the **Co**-evolution of **R**epresentation models and G**e**nerative models (**CORE**), as illustrated in Figure 1 . It aims to establish a viable evolutionary paradigm for different types of models, using data as the bridging element. Specifically, CORE is systematically divided into two stages: R2G and G2R. These stages enhance the training of generative and representation models by leveraging their interactions. Notably, these stages can form a feedback loop, allowing continuous evolution over time.

Additionally, by setting the maximization of the log-likelihood of the data $\log p(\mathbf{x})$ as our objective, we conduct a theoretical analysis (see Appendix A ). This analysis reveals that the Evidence Lower Bound (ELBO (Kingma, 2013)) of this objective can be decomposed into two distinct components: one targeting the learning of representation models and the other focusing on generative models. This decomposition further substantiates our position that both generative and representation models collaboratively learn the distribution of real-world data and can naturally assist each other.

### 3.1 R2G: FACILITATING DATA GENERATION THROUGH REPRESENTATION MODELS

Modern generative models are capable of producing diverse samples; however, they continue to face challenges with unrealistic and semantically inconsistent data generation (Abdollahzadeh et al., 2023; Yang et al., 2023). Prior research (Sun et al., 2024; Zimmermann et al., 2021) indicates that features generated by representation models retain substantial information from the original samples, suggesting a high mutual information (Shannon, 1948) between features $D_Z$ and samples $D_X$. This leads us to conjecture that representation models are naturally deconstructing complex samples into fundamental, informative, and compact "seeds" (features) that encapsulate the essential information of the original samples. We posit that these "seeds" can be reconstructed with high fidelity into their original samples, preserving high semantic information through a generative process. The comprehensive technical approach is detailed as follows.

**Training the R2G framework.** Our R2G first requires preparing a sample-feature-paired dataset $D_{XZ} = (D_X, D_Z)$ using the reference samples $D_X$:

$$(D_X, D_Z) = \{(\mathbf{x}, \mathbf{z}) \mid \mathbf{z} = \boldsymbol{f}_{\boldsymbol{\phi}}(\mathbf{x}), \mathbf{x} \in D_X\}, \tag{1}$$

where $\boldsymbol{f}_{\boldsymbol{\phi}}$ denotes a pre-trained representation model, and $\mathbf{z}$ represents the corresponding low-dimensional feature extracted from $\mathbf{x}$ using this model. Based on the constructed dataset $D_{XZ}$, the loss function for R2G can be formally described as follows:

$$\mathcal{L}_{\text{R2G}}(\boldsymbol{\theta}) = -\mathbb{E}_{(\mathbf{x},\mathbf{z})\sim(D_X,D_Z)} \left[\log p(\mathbf{x}|\boldsymbol{\theta},\mathbf{z})\right], \tag{2}$$

where $\mathcal{L}_{\text{R2G}}$ denotes the negative log-likelihood function of the generative models. As discussed in Section 2 , our investigation demonstrates that diffusion models (Ho et al., 2020) have achieved SoTA performance and garnered significant attention from the community. Therefore, we incorporate diffusion models within the R2G framework, employing their unified loss function as follows:

$$\mathcal{L}_{\text{R2G}}(\boldsymbol{\theta}) = \tfrac{1}{2}\mathbb{E}_{(\mathbf{x},\mathbf{z})\sim D_{XZ}, \boldsymbol{\epsilon}\sim\mathcal{N}(0,\mathbf{I}), \lambda\sim p(\lambda)} \left[\tfrac{w(\lambda)}{p(\lambda)} \|\hat{\boldsymbol{\epsilon}}_{\boldsymbol{\theta}}(\alpha_\lambda\mathbf{x} + \sigma_\lambda\boldsymbol{\epsilon}, \mathbf{z}; \lambda) - \boldsymbol{\epsilon}\|_2^2\right], \tag{3}$$

where the diffusion models are trained using a weighted integral of ELBOs over different noise levels (Kingma & Gao, 2024) conditioned on $\mathbf{z}$. $\lambda$ denotes the log signal-to-noise ratio associated with the timestep $t$, $w(\lambda)$ denotes the weighting function, $p(\lambda)$ denotes the distribution of $\lambda$. The predefined parameters $\alpha_\lambda$ and $\sigma_\lambda$ describe the signal and noise ratio schedules, respectively.

**Usage of representation models in the R2G framework.** While our R2G does not impose constraints on the representation models used, different representation models vary in their ability to capture image information, leading to variations in the performance of conditional generative models trained with these representations. In this study, we employ our R2G approach using self-supervised learnt representation models as the core components for dataset $D_{XZ}$, following the intuitions below:

(a) Building on prior studies (Bordes et al., 2022; Zimmermann et al., 2021; Sun et al., 2024), self-supervised learnt models capture semantic information more comprehensively than supervised learning models. This aligns with our requirements as outlined in Section 3.1.

(b) These models are particularly advantageous for application to unlabeled datasets, enabling training of $\boldsymbol{f}_{\phi}$ solely on samples $D_X$ when pre-trained models are not accessible from external sources, such as public internet repositories.

Nonetheless, we rigorously evaluate the effectiveness of different representation models within our R2G framework to enhance the training of diffusion models (see Section 4).

### 3.2 G2R: TRAINING STRONGER REPRESENTATION MODELS USING GENERATED DATA

It is well established that developing a robust representation model requires diverse and realistic training data (Radford et al., 2021). Fortunately, we can achieve high-quality data generation using the generative model framework described in Section 3.1. This naturally raises the question: *can this powerful generative model, in turn, enhance the development of a superior representation model?*

**High-quality and diverse data generation via R2G.** Utilizing the generative model developed through R2G, we can generate a synthetic dataset, $S_{XZ} = (S_X, S_Z)$, as follows:

$$S_{XZ} = (S_X, S_Z) = \{(\mathbf{x}, \mathbf{z}) \mid \mathbf{x} = \boldsymbol{g}_{\boldsymbol{\theta}}(\boldsymbol{\epsilon}, \mathbf{z}), \mathbf{z} \in D_Z, \boldsymbol{\epsilon} \sim \mathcal{N}(0, \mathbf{I})\} , \tag{4}$$

where $\boldsymbol{g}_{\boldsymbol{\theta}}(\boldsymbol{\epsilon}, \mathbf{z})$ denotes the generation process in a diffusion model, producing samples by successively obtaining $\mathbf{x}_T, \mathbf{x}_{T-1}, \ldots, \mathbf{x}_t, \mathbf{x}_0$ through:

$$\mathbf{x}_{t-1} = \frac{1}{\alpha_{\lambda_t}} \left(\mathbf{x}_t - \sigma_{\lambda_t} \hat{\boldsymbol{\epsilon}}_{\boldsymbol{\theta}}(\mathbf{x}_t, \mathbf{z}; \lambda_t)\right) + \sigma_{\lambda_{t-1}} \boldsymbol{\epsilon}_t , \tag{5}$$

where the process is initialized with $\mathbf{x}_T = \boldsymbol{\epsilon}$ and converges to $\mathbf{x}_0 = \mathbf{x}$.

**Training G2R framework through generated data.** The synthetic dataset $S_{XZ}$ serves as a knowledge carrier for the generative model, providing high-quality and diverse samples. However, these generated samples are not as realistic as the real data $D_{XZ}$, which can impair the recognition capability of a representation model trained solely on them. Consequently, $S_{XZ}$ is combined with the reference dataset $D_{XZ}$ for enhanced utility. Formally, the loss function for the G2R learning is defined as:

$$\mathcal{L}_{\text{G2R}}(\boldsymbol{\phi}) = \mathbb{E}_{(\mathbf{x}, \mathbf{z}) \sim \mathcal{I}(D_{XZ}, S_{XZ}), \boldsymbol{T} \sim \mathcal{T}} \left[\ell_{\text{ALG}}(\boldsymbol{T}(\mathbf{x}), \mathbf{z}; \boldsymbol{f}_{\boldsymbol{\phi}})\right] . \tag{6}$$

Here, $\ell_{\text{ALG}}$ denotes a generic loss function employed in representation learning algorithms[1], such as SimCLR (Chen et al., 2020). The term $\mathcal{I}(D_{XZ}, S_{XZ})$ signifies the data inflation operation applied to the two datasets, wherein these two datasets are combined into a new one. The set $\mathcal{T}$ consists of data augmentation strategies. The technical details will be elaborated subsequently.

**Practical implementations for G2R framework.** Despite the ability of generative models to produce unlimited samples and enhance representation model training data (Azizi et al., 2023), diffusion models often incur significant computational costs (Song et al., 2020). Therefore, this raises two natural questions: (a) *How can we enhance the diversity of the inflated dataset $\mathcal{I}(D_{XZ}, S_{XZ})$ without compromising its realism, given the limited synthetic data $S_{XZ}$?* (b) *How can we effectively utilize this inflated dataset?* To address these, we employ two practical techniques during training:

(a) The first technique related to $\mathcal{T}$ involves applying a milder data augmentation strategy to the inflated dataset compared to the more aggressive augmentation typically used in self-supervised learning (Chen et al., 2020). For example, a larger lower bound for RandomResizeCrop and a lower probability for color jittering can be used (see more details in Appendix C).

---

[1]Typically, only $D_X$ and $S_X$ are utilized, as most representation learning loss functions do not incorporate the use of features $\mathbf{z}$ in $D_Z$ and $S_Z$.

(b) The second technique related to $\mathcal{I}$ is reweighting, which adjusts the ratio between real and synthetic data to avoid a direct $50/50$ mixture. By increasing the proportion of real samples relative to synthetic data, the training of the representation model can be enhanced. This can be denoted as an adjustment to the ratio $\beta = (N \cdot |D|) / (N \cdot |D| + |S|)$, where $|D|$ and $|S|$ represent the number of real and synthetic data samples, respectively, and $N$ is the upweighting ratio for the real data. Please refer to the Appendix B.4 for the ablation study of these parameters.

We have conducted detailed empirical analysis for the above two techniques in Section 4.

**Efficient G2R via pseudo-supervised learning.** Most existing representation learning methods fail to fully exploit the information within inflated data $\mathcal{I}(D_{XZ}, S_{XZ})$ (refer to Footnote 1). To address this, we propose an efficient and simple approach that effectively utilizes the comprehensive information contained in data $\mathcal{I}(D_{XZ}, S_{XZ})$, leading to highly efficient model training (see Section 4). Specifically, we utilize the features $\mathbf{z}$ and corresponding samples $\mathbf{x}$ from the inflated dataset $\mathcal{I}(D_{XZ}, S_{XZ})$ to form natural pairs of samples and pseudo-targets. This approach enables pseudo-supervised learning on originally unlabeled data, thereby enhancing the efficiency of representation learning. Formally, the loss function is defined as:

$$\mathcal{L}_{\text{G2R}}(\boldsymbol{\phi}) = \frac{1}{2} \cdot \mathbb{E}_{(\mathbf{x},\mathbf{z}) \sim \mathcal{I}(D_{XZ}, S_{XZ})} \left[ \ell(\boldsymbol{f}_{\boldsymbol{\phi}}(\mathbf{x}_i), \mathbf{z}) + \ell(\boldsymbol{f}_{\boldsymbol{\phi}}(\mathbf{x}_j), \mathbf{z}) \right] , \quad (7)$$

where $\mathbf{x}_i$ and $\mathbf{x}_j$ are two augmented versions of $\mathbf{x}$, and $\ell(\mathbf{x}, \mathbf{z}) = 1 - \frac{\mathbf{x} \cdot \mathbf{z}}{\|\mathbf{x}\| \|\mathbf{z}\|}$. For implementation efficiency, it is sufficient to compute only the term $\ell(\mathbf{x}_i, \mathbf{z})$, despite a slight decrease in performance.

### 3.3 CORE: Co-evolution Loop Framework of R2G and G2R

Consequently, we propose the CORE framework (refer to panel (a) in Figure 1), which iteratively enhances the R2G and G2R processes through a feedback loop, progressively improving their performance. By initializing the co-evolution process with a lightly trained representation model and a randomly initialized generative model, we capitalize on the low cost of starting the loop from R2G.

## 4 Experiments

To assess the effectiveness of our approach, we begin by examining R2G (see Section 4.1) and G2R (see Section 4.2) individually. This initial analysis reveals how improvements in either the generative model or the representation model can positively impact the training of the other. Building on these findings, we then apply the CORE framework (see Section 4.3), evaluating its performance across datasets with varying scales and resolutions. Furthermore, to showcase the versatility of this framework, we expand our evaluation by examining its effectiveness with various self-supervised representation learning methods and different generative models (see Section 4.4).

Beyond standard experimental settings, we assess the effectiveness of CORE under unconventional and extreme conditions. One scenario involves training with limited data (see Section 4.6), which poses a substantial challenge for any model (Abdollahzadeh et al., 2023). Another critical scenario is the self-consuming loop (Shumailov et al., 2024) (see Section 4.5), where the training data for the next generation is derived from samples produced by the generative model of the previous generation. Such a loop often leads to model collapse issues in modern generative models.

### 4.1 Effectiveness Verification of R2G

**Improved representation model for training better generative model.** This experiment on the effectiveness verification of R2G is structured to investigate two problems: (a) *whether a lightly trained representation model can assist the generative model*, and (b) *how employing representation models of varying capabilities can influence the performance of generative models.* Thus, the representation model in three different capability levels are utilized: weak, moderate, and strong. These representation models are trained on the reference training dataset using a consistent training recipe, with the only variation being the number of training steps. For more details on the implementation and additional visualization analysis, please refer to Appendix C and Appendix B.1.

The experimental setup includes benchmark datasets such as CIFAR-10 (CF-10) (Krizhevsky et al., 2009), CIFAR-100 (CF-100) (Krizhevsky et al., 2009), Tiny-ImageNet (T-IN) (Le & Yang, 2015), and ImageNet-1K (IN-1K) (Deng et al., 2009). The primary metrics for evaluation are the Fréchet

Table 1: **R2G—improved representation model for better generative model.** Representation models of different capability levels, with ResNet-50 as the backbone and trained via SimCLR (Chen et al., 2020), are used to assist the generative model: ADM (Nichol & Dhariwal, 2021) for CF-10 and CF-100, and LDM (Rombach et al., 2022) for T-IN and IN-1K.

| Representation Model (used in R2G) | Generative Model | | | | | | | |
|---|---|---|---|---|---|---|---|---|
| | FID↓ | | | | IS↑ | | | |
| | CF-10 | CF-100 | T-IN | IN-1K | CF-10 | CF-100 | T-IN | IN-1K |
| Baseline (Unconditional) | 5.34 | 8.30 | 17.99 | 43.40 | 8.99 | 10.35 | 12.73 | 22.19 |
| Weak Level | 4.84 (-0.50) | 7.86 (-0.44) | 16.63 (-1.36) | 38.04 (-5.36) | 9.14 (+0.15) | 10.19 (-0.16) | 12.90 (+0.17) | 25.71 (+3.52) |
| Moderate Level | 3.82 (-1.52) | 5.58 (-2.72) | 13.35 (-4.64) | 26.28 (-17.12) | 9.45 (+0.46) | 10.91 (+0.56) | 14.14 (+1.41) | 39.42 (+17.23) |
| Strong Level | 3.41 (-1.93) | 4.93 (-3.37) | 10.32 (-7.67) | 20.13 (-23.27) | 9.58 (+0.59) | 11.33 (+0.98) | 16.22 (+3.49) | 50.47 (+28.28) |

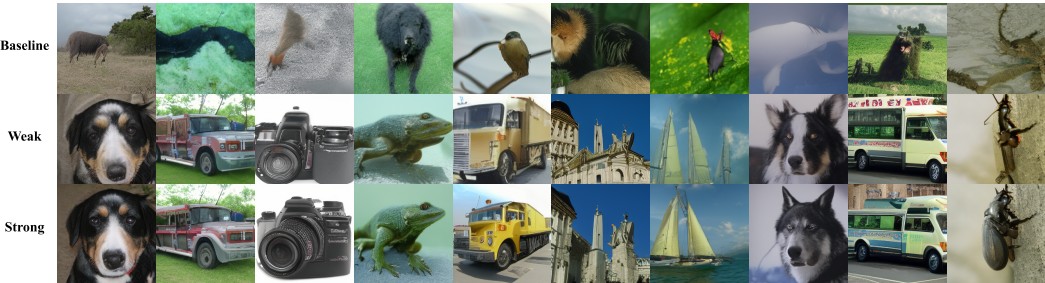

Figure 2: **Visualization of generated images of R2G vs. baseline model over ImageNet-1K.** For our R2G, we utilize both weak and strong representation models during the training phase.

Inception Distance (FID) (Heusel et al., 2017) and Inception Score (IS) (Salimans et al., 2016), which are widely recognized measures for evaluating the quality and diversity of generated images.

As shown in Table 1, it is evident that even a minimally trained representation model can enhance the generative model, resulting in improved FID performance compared to the naive baseline[2]. As the representation model becomes more sophisticated, we observe corresponding improvements in the generative model across datasets. These findings indicate: (a) implementing R2G is nearly cost-free, and (b) enhanced representation models can yield superior generative models through R2G.

Furthermore, Figure 2 visually demonstrates that applying R2G can bring more realistic semantic information to generated images compared to the baseline model.

### 4.2 Effectiveness Verification of G2R

**Enhanced generative models for training superior representation Models** This experiment investigates how advancements in generative models can improve the performance of self-supervised representation models via generated samples and our G2R approach. Specifically, it addresses the question:

*Can a newly improved generative model, developed using R2G, aid in the training of the representation model?*

To explore the impact of generative models on representation learning, we introduce datasets generated by models with varying levels of generative capability. As a result, all representation models previously used as "assistant models" for R2G now function as the "to-be-assisted models" for G2R. Specifically, in this section, we derive three generative

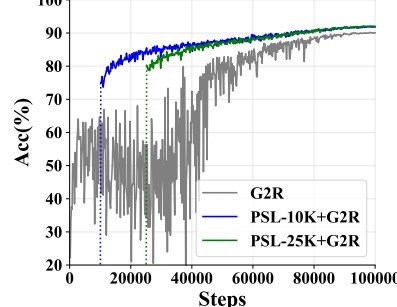

Figure 3: **Efficient G2R training via pseudo-supervised learning (PSL).** PSL-10K+G2R indicates that PSL is applied during the first 10K steps of training, with the overall training cost remaining unchanged.

models using representation models categorized as weak, moderate, and strong through R2G. These generative models are continuously referenced at these levels (see the first column of Table 2). Additionally, our experiment assesses the offline linear probe accuracy (Chen et al., 2020) of the representation model, evaluating its performance in downstream classification tasks.

Table 2 demonstrates that as the generative model's capabilities and the quality of its generated images improve, there is a corresponding increase in the linear probing accuracy of the representation

---

[2]The naive baseline refers to the method that does not employ G2R for developing generative models and relies on unconditional training and sampling.

Table 2: **G2R—improved generative model for training better representation model.** The generative model derived from R2G, as shown in Table 1, aids in training the representation model by sampling synthetic data. Linear probing accuracy is presented for both the baseline and its enhanced version utilizing G2R.

| Generative Model (with different R2G stage) | CF-10 | CF-100 | T-IN | IN-1K |
|---|---|---|---|---|
| Baseline | $89.5 \pm 0.1$ | $66.3 \pm 0.1$ | $43.8 \pm 0.2$ | $58.0 \pm 0.0$ |
| Weak | $89.4 \pm 0.1$ (-0.1) | $65.0 \pm 0.1$ (-1.3) | $44.2 \pm 0.2$ (+0.3) | $58.0 \pm 0.0$ (+0.0) |
| Moderate | $90.5 \pm 0.1$ (+1.0) | $66.6 \pm 0.2$ (+0.3) | $44.8 \pm 0.5$ (+1.0) | $58.1 \pm 0.1$ (+0.1) |
| Strong | $90.9 \pm 0.0$ (+1.4) | $67.9 \pm 0.0$ (+2.9) | $45.9 \pm 0.2$ (+2.1) | $58.4 \pm 0.1$ (+0.4) |

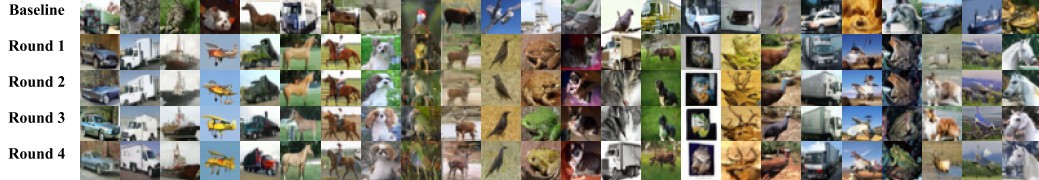

Figure 4: **Visualization of generated images of CORE vs. baseline model over CIFAR-10.** For our CORE, we conduct training using multiple co-evolution rounds. During these rounds, images are generated by progressively evolved generative models.

model. When conducting G2R at moderate and strong levels, the representation model consistently outperforms the baseline across all datasets, confirming the effectiveness of G2R in enhancing representation models. Additionally, we can enhance our G2R by applying pseudo-supervised learning (PSL) to achieve more efficient training. At the same training cost, beginning with PSL for an initial phase before transitioning to the original G2R leads to a more efficient learning process compared to the default G2R, as shown in Figure 3 . For more results of PSL refer to Appendix B.5 .

These results suggest the potential for mutual enhancement between generative and representation models, therefore establishing a closed loop of R2G and G2R to promote ongoing advancements in both model types. Based on this loop, we delve into the multi-round evolutionary process, CORE, in the subsequent subsections.

### 4.3 CO-EVOLUTION FRAMEWORK ON DATASETS OF DIFFERENT RESOLUTIONS AND SCALES

**Implementations on CIFAR-10/CIFAR-100.** In our experiments on the CIFAR-10 and CIFAR-100 datasets, we employ the CORE framework, utilizing ADM (Nichol & Dhariwal, 2021) as the generative model and a ResNet-50 trained with the SimCLR (Chen et al., 2020) method as the representation model within the co-evolution loop.

For the implementation of ADM, we utilize residual blocks for both up-sampling and down-sampling, a straightforward technique shown to enhance performance as indicated in the guided diffusion literature (Dhariwal & Nichol, 2021; Song et al., 2021). All other settings are maintained in alignment with the original ADM model, including 1,000 diffusion steps, a base network width of 128, and attention mechanisms at resolutions of 16 and 8.

In the initial phase, we train ResNet-50 exclusively with the original training dataset, omitting any synthetic data. During subsequent iterations of the co-evolution process with G2R, we incorporate synthetic data produced by the generative model and retrain the representation model from scratch. The projector's hidden dimension is set to 2,048, with an output dimension of 128. This output is then utilized by the generative model for R2G. During each G2R phase of the co-evolution loop, the generative model produces 100,000 samples.

**Implementations on Tiny-ImageNet.** For our experiments using CORE on the Tiny-ImageNet dataset, we employed the Latent Diffusion Model (LDM) as our generative model (Rombach et al., 2022). The encoder and decoder in the LDM were adapted from the open-source VQGAN framework (Esser et al., 2021), utilizing a latent space of 4-channel encodings at a resolution of 32. This configuration enhances training efficiency on large-scale, high-resolution image datasets compared to pixel-based diffusion models. The representation model used is a ResNet-50 trained with SimCLR, where the projector's hidden dimension is expanded to 4,096 and the output dimension increased to 512. For each G2R, we generated 100K synthetic images.

**Analysis of experimental results.** The experimental results presented in Figure 5 indicate that *the proposed algorithm can consistently improve the performance of generative and representation*

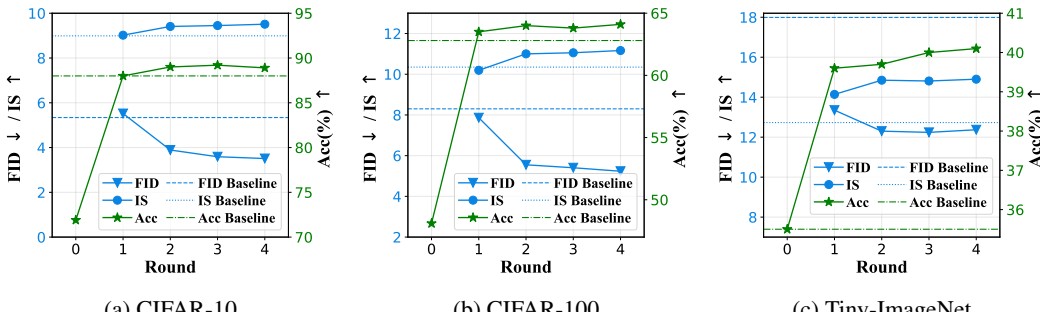

(a) CIFAR-10      (b) CIFAR-100      (c) Tiny-ImageNet

Figure 5: **Evaluation of our CORE over three datasets**. We assess the performance of both generative and representation models as the evolution rounds of CORE progress.

Table 3: **Evaluation of Representation Learning Methods with and without CORE**. This study compares four self-supervised learning techniques—SimCLR, MoCo, BYOL, and Barlow Twins—against supervised learning models for representation learning. Furthermore, the performance of a self-supervised model trained on CIFAR-10 and transferred to Tiny-ImageNet is assessed, as shown in the SimCLR* row.

| | Generative Model | | | | Representation Model | |
|---|---|---|---|---|---|---|
| Representation Learning Method | FID↓ | | IS↑ | | Acc(%)↑ | |
| | Baseline | w/ R2G | Baseline | w/ R2G | Baseline | w/ G2R |
| SimCLR | | 3.41 (-1.93) | | 9.48 (+0.49) | 89.5±0.1 | 90.9±0.0 (+1.4) |
| MoCo | | 3.34 (-2.00) | | 9.55 (+0.56) | 91.1±0.0 | 92.1±0.0 (+1.0) |
| BYOL | 5.34 | 3.20 (-2.14) | 8.99 | 9.62 (+0.63) | 91.3±0.1 | 90.8±0.1 (-0.5) |
| Barlow Twins | | 4.73 (-0.61) | | 9.25 (+0.26) | 88.5±0.3 | 89.7±0.2 (+1.2) |
| Supervised Learning | | 4.95 (-0.39) | | 9.17 (+0.18) | 92.0±0.0 | 93.6±0.0 (+1.6) |
| SimCLR* (CF10→T-IN) | 17.99 | 17.52 (-0.47) | 12.73 | 13.34 (+0.61) | 43.8±0.2 | 44.5±0.3 (+0.7) |

*models as the number of evolutionary rounds increases.* Furthermore, Figure 4 shows that our algorithm's generated images exhibit more explicit semantic information compared to the baseline.

### 4.4 Experiments with Intensive Generative and Representation Models

This subsection examines various representation learning techniques and generative models to assess performance across different settings. **G2R:** For representation learning, we apply four prevalent self-supervised learning (SSL) methods: SimCLR (Chen et al., 2020), MoCo (He et al., 2020), BYOL (Grill et al., 2020), and Barlow Twins (Zbontar et al., 2021). Each of these approaches enables representation learning without reliance on labeled data. ResNet-50 serves as the backbone architecture, and ADM is employed as the generative model for all four SSL methods. Additionally, we incorporate a supervised learning approach utilizing the cross-entropy loss function.

**R2G:** To further explore generative modeling, we investigate three distinct methods: ADM (Nichol & Dhariwal, 2021), LDM (Rombach et al., 2022), and DiT (Peebles & Xie, 2023). For consistency, we employ a ResNet-50 model trained with SimCLR as the representation model across all three types of generative models mentioned above.

Table 3 and Table 4 showcase the effectiveness of our CORE across different representation and generative learning methods on various datasets, outperforming multiple baseline algorithms.

### 4.5 Use CORE to Help Self-consuming Loop

Self-Consuming Loop (Alemohammad et al., 2024) training refers to the cyclic process in generative model training where synthetic data generated by earlier generations is used as training data for subsequent generations. Studying this phenomenon is crucial, as the growing prevalence of synthetic data on the internet and within standard datasets suggests that future models will likely rely, at least in part, on a mix of real and synthetic data, thereby forming a self-consuming loop. In the absence of sufficient fresh real data, such a loop can lead to a degradation in the quality (precision) or diversity (recall) of generative models across generations, which is known as Model Autophagy Disorder (MAD) (Alemohammad et al., 2024; Bertrand et al., 2024). Self-consuming loop training can be categorized into several types (Alemohammad et al., 2024): *Fully Synthetic Loop*, *Synthetic Augmentation Loop*, and *Fresh Data Loop*.

Table 4: **Comparison of generative models with and without CORE**. The study includes the following prominent generative models: ADM, LDM, and DiT.

| Generative Model | Generative Model | | | | Representation Model | |
|---|---|---|---|---|---|---|
| | FID↓ | | IS↑ | | Acc(%)↑ | |
| | Baseline | w/ R2G | Baseline | w/ R2G | Baseline | w/ G2R |
| ADM | 24.77 | 17.39 (-7.38) | 11.85 | 14.41 (+2.56) | | 44.2±0.1 (+0.4) |
| LDM | 17.99 | 10.32 (-7.67) | 12.73 | 16.22 (+3.49) | 43.8± 0.2 | 45.9±0.2 (+2.1) |
| DiT | 26.44 | 13.27 (-13.17) | 10.65 | 15.25 (+4.60) | | 45.0±0.1 (+1.3) |

To validate the potential of our proposed CORE framework in addressing the challenges posed by self-consuming loop training, we conducted experiments in the particularly demanding conditions: the Synthetic Augmentation Loop. In this scenario, the training data is not only drawn from the original reference training dataset but also sampled from the previously trained generative model. Specifically, for CIFAR-10 as the reference training dataset, we use the previous-generation generative model to create a class-balanced set of 50,000 images, which serves as the half of the new dataset for training the next-generation generative model. Here, we utilize CORE to prevent self-consuming loop training from entering "MAD". In addition to the previously discussed adjustments to the generative model's training dataset, we maintain the use of a mixture of the reference real dataset and the synthetic dataset for G2R.

## 4.6 CORE MITIGATES DATA-SCARCE SCENARIOS

Data-scarce scenarios refers to situations where only a limited amount of labeled or unlabeled data is available for training machine learning models (Bansal et al., 2022). This situation is common in domains where data collection is expensive, time-consuming, or impractical due to privacy concerns, regulatory restrictions, or the rarity of specific phenomena. For both representation models and generative models, having fewer data points makes it challenging to

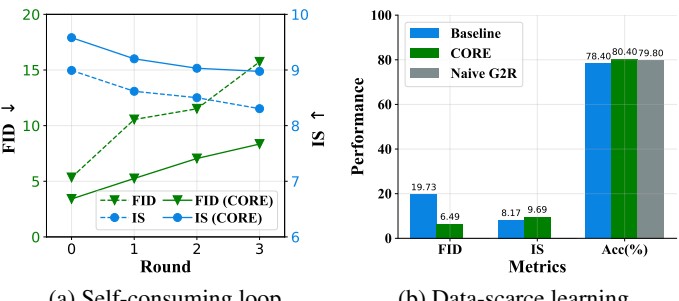

(a) Self-consuming loop  (b) Data-scarce learning

Figure 6: **CORE works effectively in different scenarios.** We are conducting two experiments: self-consuming loop training and data-scarce learning, both of which make certain negative assumptions about the training dataset, making it more challenging for the model to learn.

learn rich, informative features. This limitation impairs the model's ability to capture the underlying structure of the data, leading to suboptimal performance in tasks (Abdollahzadeh et al., 2023).

To simulate the data-scarce learning setting, we use only one-tenth of the CIFAR-10 training set as our initial training dataset. Using the default setup, which includes a ResNet-50 trained via SimCLR as the representation model and ADM as the generative model. Figure 6b demonstrates that our CORE can effectively assist both generative and representation models to achieve superior performance in this scenario.

## 5 LIMITATION AND CONCLUSION

We present CORE, a preliminary attempt to explore the co-evolution learning of generative and representation models. It consists of two interdependent frameworks, R2G and G2R, which together create a closed-loop system. The semantic information extracted by the representation model facilitates the training of the generative model, while the generative model, in turn, enhances the representation model by synthesizing new data. We empirically validate the effectiveness of both G2R and R2G, and evaluate the performance of CORE across various datasets. We further demonstrate its potential through two scenarios: the self-consuming loop and data-scarce learning. However, the conditions necessary for effectively launching CORE and how to efficiently approach its performance limits with minimal training costs require more in-depth theoretical research and potentially larger-scale experimental exploration. Nonetheless, we hope that our current progress can inspire researchers to investigate the potential of co-evolution between two or even more models, and perhaps provide valuable insights.

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

## A  THEORETICAL ANALYSIS OF CORE

**Problem Setup.**

(a) **Dataset:** $\mathcal{X} = \{\mathbf{x}^{(1)}, \mathbf{x}^{(2)}, \ldots, \mathbf{x}^{(N)}\}$, where each $\mathbf{x}^{(i)}$ is generated from a joint latent variable model.

(b) **Latent variables:** $\mathbf{z}$: A representation distribution to be learned from the data. $\boldsymbol{\epsilon}$: A latent variable with a known conditional distribution $p(\boldsymbol{\epsilon}|\mathbf{z})$ given $\mathbf{z}$.

(c) **Objective:** Develop an ELBO that enables learning both a good representation $p(\mathbf{z})$ and a robust generative model $p(\mathbf{x}|\mathbf{z}, \boldsymbol{\epsilon})$ by maximizing this ELBO.

**Evidence lower bound (ELBO).**   For a single data point $\mathbf{x}$, the marginal likelihood is:

$$p(\mathbf{x}) = \int \int p(\mathbf{x}, \mathbf{z}, \boldsymbol{\epsilon}) \, d\mathbf{z} \, d\boldsymbol{\epsilon} = \int \int p(\mathbf{x}|\mathbf{z}, \boldsymbol{\epsilon}) p(\boldsymbol{\epsilon}|\mathbf{z}) p(\mathbf{z}) \, d\mathbf{z} \, d\boldsymbol{\epsilon} \tag{8}$$

To derive the ELBO, we introduce $q(\mathbf{z}, \boldsymbol{\epsilon}|\mathbf{x})$, an approximation to the true posterior $p(\mathbf{z}, \boldsymbol{\epsilon}|\mathbf{x})$. Starting with the log marginal likelihood:

$$\log p(\mathbf{x}) = \log \int \int p(\mathbf{x}, \mathbf{z}, \boldsymbol{\epsilon}) \, d\mathbf{z} \, d\boldsymbol{\epsilon} \tag{9}$$

Insert the variational posterior $q(\mathbf{z}, \boldsymbol{\epsilon}|\mathbf{x})$ by multiplying and dividing inside the integral:

$$\log p(\mathbf{x}) = \log \int \int q(\mathbf{z}, \boldsymbol{\epsilon}|\mathbf{x}) \frac{p(\mathbf{x}, \mathbf{z}, \boldsymbol{\epsilon})}{q(\mathbf{z}, \boldsymbol{\epsilon}|\mathbf{x})} \, d\mathbf{z} \, d\boldsymbol{\epsilon} \tag{10}$$

Apply Jensen's inequality (due to the concavity of the logarithm):

$$\log p(\mathbf{x}) \geq \int \int q(\mathbf{z}, \boldsymbol{\epsilon}|\mathbf{x}) \log \frac{p(\mathbf{x}, \mathbf{z}, \boldsymbol{\epsilon})}{q(\mathbf{z}, \boldsymbol{\epsilon}|\mathbf{x})} \, d\mathbf{z} \, d\boldsymbol{\epsilon} = \mathcal{L}(\mathbf{x}) \tag{11}$$

where $\mathcal{L}(\mathbf{x})$ is the ELBO. The ELBO can be expressed as:

$$\mathcal{L}(\mathbf{x}) = \mathbb{E}_{q(\mathbf{z},\boldsymbol{\epsilon}|\mathbf{x})}[\log p(\mathbf{x}, \mathbf{z}, \boldsymbol{\epsilon})] - \mathbb{E}_{q(\mathbf{z},\boldsymbol{\epsilon}|\mathbf{x})}[\log q(\mathbf{z}, \boldsymbol{\epsilon}|\mathbf{x})] \tag{12}$$

Expanding $p(\mathbf{x}, \mathbf{z}, \boldsymbol{\epsilon})$:

$$\mathcal{L}(\mathbf{x}) = \mathbb{E}_{q(\mathbf{z},\boldsymbol{\epsilon}|\mathbf{x})}[\log p(\mathbf{x}|\mathbf{z}, \boldsymbol{\epsilon}) + \log p(\boldsymbol{\epsilon}|\mathbf{z}) + \log p(\mathbf{z})] - \mathbb{E}_{q(\mathbf{z},\boldsymbol{\epsilon}|\mathbf{x})}[\log q(\mathbf{z}, \boldsymbol{\epsilon}|\mathbf{x})] \tag{13}$$

Rearranging terms:

$$\mathcal{L}(\mathbf{x}) = \mathbb{E}_{q(\mathbf{z},\boldsymbol{\epsilon}|\mathbf{x})}[\log p(\mathbf{x}|\mathbf{z}, \boldsymbol{\epsilon})] + \mathbb{E}_{q(\mathbf{z},\boldsymbol{\epsilon}|\mathbf{x})}[\log p(\boldsymbol{\epsilon}|\mathbf{z}) + \log p(\mathbf{z}) - \log q(\mathbf{z}, \boldsymbol{\epsilon}|\mathbf{x})] \tag{14}$$

Recognize that the second term is the negative Kullback-Leibler (KL) divergence between $q(\mathbf{z}, \boldsymbol{\epsilon}|\mathbf{x})$ and $p(\mathbf{z}, \boldsymbol{\epsilon})$:

$$\mathcal{L}(\mathbf{x}) = \mathbb{E}_{q(\mathbf{z},\boldsymbol{\epsilon}|\mathbf{x})}[\log p(\mathbf{x}|\mathbf{z}, \boldsymbol{\epsilon})] - \mathrm{KL}(q(\mathbf{z}, \boldsymbol{\epsilon}|\mathbf{x}) \| p(\mathbf{z}, \boldsymbol{\epsilon})) \tag{15}$$

Given that $p(\mathbf{z}, \boldsymbol{\epsilon}) = p(\boldsymbol{\epsilon}|\mathbf{z}) p(\mathbf{z})$, we have:

$$\mathcal{L}(\mathbf{x}) = \mathbb{E}_{q(\mathbf{z},\boldsymbol{\epsilon}|\mathbf{x})}[\log p(\mathbf{x}|\mathbf{z}, \boldsymbol{\epsilon})] - \mathrm{KL}(q(\mathbf{z}, \boldsymbol{\epsilon}|\mathbf{x}) \| p(\boldsymbol{\epsilon}|\mathbf{z}) p(\mathbf{z})) \tag{16}$$

**Simplifying the ELBO.**   To make the ELBO tractable and implementable, we make specific assumptions about the structure of the variational posterior $q(\mathbf{z}, \boldsymbol{\epsilon}|\mathbf{x})$. A common and practical choice is to assume that $\boldsymbol{\epsilon}$ is conditionally independent of $\mathbf{z}$ given $\mathbf{x}$, or to model certain dependencies to reflect the generative process. However, to align closely with the generative model structure $p(\boldsymbol{\epsilon}|\mathbf{z})$, a sensible assumption is:

$$q(\mathbf{z}, \boldsymbol{\epsilon}|\mathbf{x}) = q(\boldsymbol{\epsilon}|\mathbf{z}, \mathbf{x}) q(\mathbf{z}|\mathbf{x}) \tag{17}$$

However, to further simplify, we might assume that $\boldsymbol{\epsilon}$ depends only on $\mathbf{z}$, reflecting the generative model's dependency, leading to:

$$q(\mathbf{z}, \boldsymbol{\epsilon}|\mathbf{x}) = q(\boldsymbol{\epsilon}|\mathbf{z}) q(\mathbf{z}|\mathbf{x}) \tag{18}$$

This approximation assumes that given $\mathbf{z}$, the posterior over $\boldsymbol{\epsilon}$ does not directly depend on $\mathbf{x}$, which aligns with $p(\boldsymbol{\epsilon}|\mathbf{z})$ being known and independent of $\mathbf{x}$.

**Note:** The choice of variational posterior structure depends on the specific model and desired tractability. Here, we choose $q(\mathbf{z}, \boldsymbol{\epsilon}|\mathbf{x}) = q(\boldsymbol{\epsilon}|\mathbf{z})q(\mathbf{z}|\mathbf{x})$ to align with the generative model and maintain tractability. Given the factorization $q(\mathbf{z}, \boldsymbol{\epsilon}|\mathbf{x}) = q(\boldsymbol{\epsilon}|\mathbf{z})q(\mathbf{z}|\mathbf{x})$, the KL divergence becomes:

$$\mathrm{KL}(q(\mathbf{z}, \boldsymbol{\epsilon}|\mathbf{x})\|p(\boldsymbol{\epsilon}|\mathbf{z})p(\mathbf{z})) = \mathrm{KL}(q(\mathbf{z}|\mathbf{x})q(\boldsymbol{\epsilon}|\mathbf{z})\|p(\boldsymbol{\epsilon}|\mathbf{z})p(\mathbf{z})) \tag{19}$$

Using the property that KL divergence is additive over independent distributions:

$$\mathrm{KL}(q(\mathbf{z}|\mathbf{x})q(\boldsymbol{\epsilon}|\mathbf{z})\|p(\boldsymbol{\epsilon}|\mathbf{z})p(\mathbf{z})) = \mathrm{KL}(q(\mathbf{z}|\mathbf{x})\|p(\mathbf{z})) + \mathbb{E}_{q(\mathbf{z}|\mathbf{x})}[\mathrm{KL}(q(\boldsymbol{\epsilon}|\mathbf{z})\|p(\boldsymbol{\epsilon}|\mathbf{z}))] \tag{20}$$

Substituting the decomposition back into the ELBO:

$$\mathcal{L}(\mathbf{x}) = \mathbb{E}_{q(\mathbf{z}|\mathbf{x})q(\boldsymbol{\epsilon}|\mathbf{z})}[\log p(\mathbf{x}|\mathbf{z}, \boldsymbol{\epsilon})] - \mathrm{KL}(q(\mathbf{z}|\mathbf{x})\|p(\mathbf{z})) - \mathbb{E}_{q(\mathbf{z}|\mathbf{x})}[\mathrm{KL}(q(\boldsymbol{\epsilon}|\mathbf{z})\|p(\boldsymbol{\epsilon}|\mathbf{z}))] \tag{21}$$

Alternatively, recognizing that $\mathbb{E}_{q(\mathbf{z}|\mathbf{x})}[\mathrm{KL}(q(\boldsymbol{\epsilon}|\mathbf{z})\|p(\boldsymbol{\epsilon}|\mathbf{z}))]$ is an expectation over $\mathbf{z}$, it can be written as:

$$\mathcal{L}(\mathbf{x}) = \mathbb{E}_{q(\mathbf{z}|\mathbf{x})q(\boldsymbol{\epsilon}|\mathbf{z})}[\log p(\mathbf{x}|\mathbf{z}, \boldsymbol{\epsilon})] - \mathrm{KL}(q(\mathbf{z}|\mathbf{x})\|p(\mathbf{z})) - \mathbb{E}_{q(\mathbf{z}|\mathbf{x})}[\mathrm{KL}(q(\boldsymbol{\epsilon}|\mathbf{z})\|p(\boldsymbol{\epsilon}|\mathbf{z}))] \tag{22}$$

**Final ELBO expression.** Consolidating the above derivations, the final ELBO suitable for our model is:

$$\mathcal{L}(\mathbf{x}) = \mathbb{E}_{q(\mathbf{z}|\mathbf{x})q(\boldsymbol{\epsilon}|\mathbf{z})}[\log p(\mathbf{x}|\mathbf{z}, \boldsymbol{\epsilon})] - \mathrm{KL}(q(\mathbf{z}|\mathbf{x})\|p(\mathbf{z})) - \mathbb{E}_{q(\mathbf{z}|\mathbf{x})}[\mathrm{KL}(q(\boldsymbol{\epsilon}|\mathbf{z})\|p(\boldsymbol{\epsilon}|\mathbf{z}))] \tag{23}$$

Alternatively, if we assume that $q(\boldsymbol{\epsilon}|\mathbf{z}) = p(\boldsymbol{\epsilon}|\mathbf{z})$ (i.e., the posterior $q(\boldsymbol{\epsilon}|\mathbf{z})$ matches the prior $p(\boldsymbol{\epsilon}|\mathbf{z})$), then the last KL term disappears, simplifying the ELBO to:

$$\mathcal{L}(\mathbf{x}) = \underbrace{\mathbb{E}_{q(\mathbf{z}|\mathbf{x})p(\boldsymbol{\epsilon}|\mathbf{z})}[\log p(\mathbf{x}|\mathbf{z}, \boldsymbol{\epsilon})]}_{\text{Generative}} - \underbrace{\mathrm{KL}(q(\mathbf{z}|\mathbf{x})|p(\mathbf{z}))}_{\text{Representation}} \tag{24}$$

However, this simplification assumes that $q(\boldsymbol{\epsilon}|\mathbf{z})$ perfectly matches $p(\boldsymbol{\epsilon}|\mathbf{z})$, which may not always hold. Therefore, unless such an assumption is justified, the more general form with the expectation over the KL divergence should be used.

**Optimization and learning.** By maximizing the ELBO across all data points, we achieve:

(a) **Representation Learning ($p(\mathbf{z})$):** The term $\mathrm{KL}(q(\mathbf{z}|\mathbf{x})\|p(\mathbf{z}))$ guides the learning of $p(\mathbf{z})$ to be a meaningful prior that captures the underlying structure of the data.

(b) **Generative Modeling ($p(\mathbf{x}|\mathbf{z}, \boldsymbol{\epsilon})$):** The expected log-likelihood $\mathbb{E}[\log p(\mathbf{x}|\mathbf{z}, \boldsymbol{\epsilon})]$ ensures that the generative model can accurately produce/reconstruct data points from the latent variables.

(c) **Prior Alignment ($p(\boldsymbol{\epsilon}|\mathbf{z})$):** The expectation $\mathbb{E}[\mathrm{KL}(q(\boldsymbol{\epsilon}|\mathbf{z})\|p(\boldsymbol{\epsilon}|\mathbf{z}))]$ ensures that the inferred $\boldsymbol{\epsilon}$ given $\mathbf{z}$ follows the known conditional distribution, integrating prior knowledge into the model.

**Summary.** *In this paper, our proposed* CORE *naturally conducts temporally asynchronous learning for the two terms in* (24).

# B ADDITIONAL RESULTS

## B.1 ADDITIONAL RESULTS FOR R2G

Owing to the representation model's ability to extract semantic information, the generative model after R2G demonstrates strong semantic consistency in its generated outputs, as shown in Figure 7 and Figure 8. The top row in the figures consists of the original real images, which are used as inputs to the representation model to obtain the representation vectors. The other images in each column are generated based on the same reference image with different randomly initialized noise. It can be seen from each column of images (especially the generated results for ImageNet-1K at $256 \times 256$) that the generative model captures the most important semantic information from the original images, while also exhibiting diversity in aspects such as color and composition. These results demonstrate that the generated images effectively retain semantic information.

As mentioned in Section 3, G2R is rooted in the intuition that we can utilize the features obtained from the representation model that encapsulate the essential information of the original samples. Therefore, our generative model training essentially focuses on finding the connection between this representation space (which, in our case, consists of 128 or 512 dimensions) and the image space. This connection can be more easily optimized compared to solely starting from Gaussian noise.

Moreover, if these synthetic images are used for training the representation model in G2R, it would be akin to a "realistic" data augmentation operation, which is crucial for self-supervised learning.

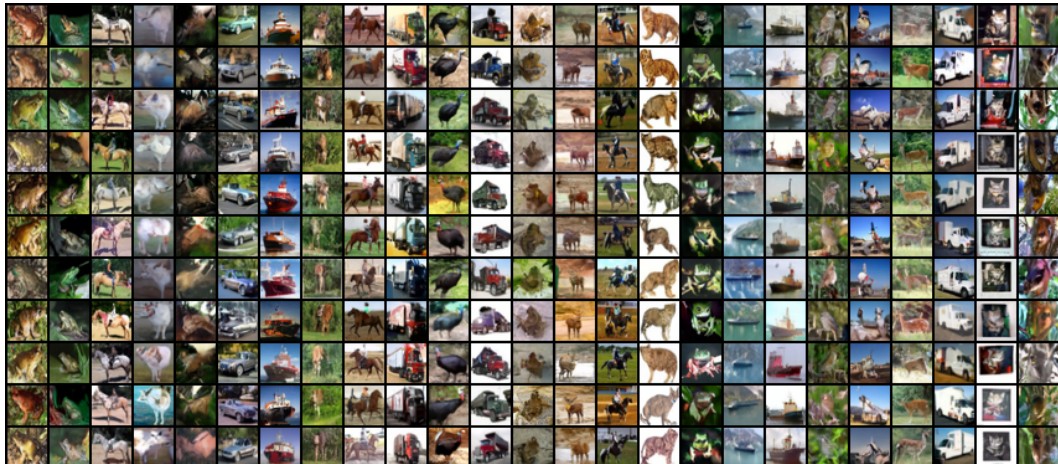

Figure 7: **The images generated by the generative model after R2G training exhibit good semantic consistency. (CIFAR-10 32×32)** Each column in the figure represents different generation results for the same representation vector, with the top row being the original images.

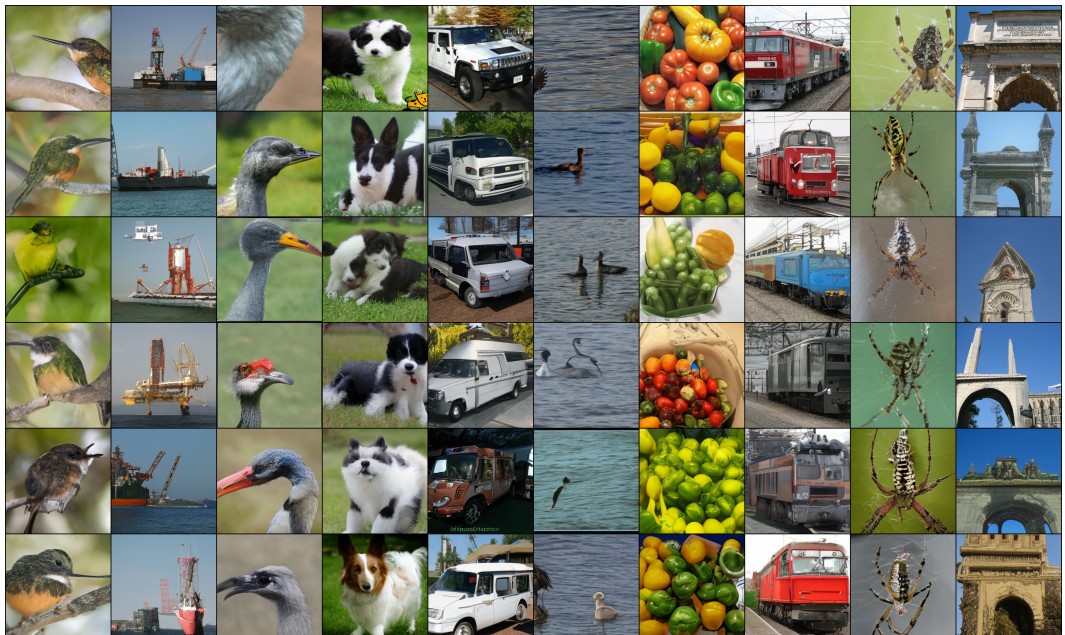

Figure 8: **The images generated by the generative model after R2G training exhibit good semantic consistency. (ImageNet-1K 256×256)** Each column in the figure represents different generation results for the same representation vector, with the top row being the original images.

## B.2 ADDITIONAL RESULTS FOR G2R

In Section 4.2 , we utilize generative models corresponding to the weak, moderate, and strong levels for G2R. Here, we present the performance differences of the representation model before and after G2R given the same training budget, as shown in Table 5 . It is observed that after using G2R, the representation model demonstrates improvements across all datasets, irrespective of whether it operates at the weak, moderate, or strong level. This result provides additional evidence for the effectiveness of G2R.

Table 5: **G2R—improved generative model for better representation model.** The generative model obtained from R2G in Table 1 can assist in training the representation model by sampling synthetic data. The linear probing accuracy is reported both before G2R and after G2R, while maintaining the same training cost.

| Generative Model (with different R2G stage) | Representation Model ↑ | | | | | | | |
|---|---|---|---|---|---|---|---|---|
| | Acc(%) Before G2R | | | | Acc(%) After G2R | | | |
| | CF-10 | CF-100 | T-IN | IN-1K | CF-10 | CF-100 | T-IN | IN-1K |
| Weak | 77.5 | 48.1 | 24.9 | 37.2 | 80.6 (+3.1) | 52.1 (+4.0) | 27.9 (+3.0) | 38.7 (+1.5) |
| Moderate | 87.6 | 63.2 | 35.3 | 55.0 | 88.1 (+0.5) | 63.6 (+0.4) | 39.6 (+4.3) | 55.1 (+0.1) |
| Strong | 89.5 | 66.3 | 43.8 | 58.0 | 90.9 (+1.4) | 67.9 (+1.6) | 45.9 (+2.1) | 58.4 (+0.4) |

## B.3 ADDITIONAL RESULTS FOR CORE OF DIFFERENT RESOLUTIONS AND SCALES

We use CORE to perform experiments on multiple datasets of varying resolutions and scales, validating its effectiveness, as shown in Figure 5. Here, we provide specific values to compare the performance improvements of the generative model or representation model when using CORE versus not using CORE, as shown in Table 6. The results from the table indicate that CORE achieves notable performance improvements across different datasets compared to the baseline.

Table 6: **Performance comparison of generative and representation models with CORE.** The "w/ CORE" column summarizes model performance in the final evaluation, while the baseline results, representing models trained exclusively on real data, are displayed in the "w/o CORE" column.

| Dataset | Generative Model | | | | Representation Model | |
|---|---|---|---|---|---|---|
| | FID ↓ | | IS ↑ | | Acc(%) ↑ | |
| | w/o CORE | w/ CORE | w/o CORE | w/ CORE | w/o CORE | w/ CORE |
| CF-10 | 5.34 | 3.51 (-1.83) | 8.99 | 9.51 (+0.52) | $71.9 \pm 0.4$ | $89.2 \pm 0.0$ (+17.3) |
| CF-100 | 8.30 | 5.24 (-3.06) | 10.35 | 11.17 (+0.82) | $62.8 \pm 0.0$ | $64.1 \pm 0.3$ (+1.3) |
| T-IN | 17.99 | 12.29 (-5.70) | 12.73 | 14.81 (+2.08) | $35.5 \pm 0.3$ | $40.1 \pm 0.3$ (+4.6) |

## B.4 ABLATION EXPERIMENTS ON THE REWEIGHTING RATIO AND THE NUMBER OF SYNTHETIC IMAGES

The synthetic dataset contains 100K images and the reweighting ratio is set to 2 as the default setting in our experiments (for both CIFAR-10/100 and Tiny-ImageNet). Different scales of synthetic datasets and mixing ratios result in varying performance levels (Wang et al., 2024b). Therefore, we examine the effects of synthetic dataset size and reweighting ratio on T-IN. As illustrated in Table 7, the best performance is achieved at different reweighting ratios depending on the scale of the synthetic data. The results show that when adding more synthetic data, higher reweighting ratio should be applied to achieve the best performance.

Table 7: **Ablation study on synthetic data scale and reweighting ratio on T-IN.** We use the generative model obtained from R2G, as mentioned in Table 1. The linear probing accuracy is reported while ensuring that the training costs are consistent across all synthetic data scales and reweighting ratios.

| Synthetic Data Scale | Reweighting Ratio | | | |
|---|---|---|---|---|
| | 1 | 2 | 5 | 10 |
| 0 (Baseline) | $43.8 \pm 0.2$ | | | |
| 100K | $45.1 \pm 0.2$ | $\mathbf{45.9 \pm 0.2}$ | $45.8 \pm 0.1$ | $45.6 \pm 0.2$ |
| 500K | $45.0 \pm 0.2$ | $45.1 \pm 0.2$ | $45.3 \pm 0.2$ | $\mathbf{45.5 \pm 0.2}$ |
| 1M | $44.6 \pm 0.2$ | $45.3 \pm 0.2$ | $45.4 \pm 0.1$ | $\mathbf{45.8 \pm 0.1}$ |

## B.5 ADDITIONAL RESULTS FOR PSEUDO-SUPERVISED LEARNING IN G2R.

Due to the natural construction of data-label pairs $(\mathbf{x}, \mathbf{z})$ during the G2R in CORE, it facilitates the possibility of pseudo-supervised learning (PSL). As shown in Figure 3, employing PSL aids in training the representation model more effectively. Here, we present more detailed results, as shown in Table 8, which provides the linear probing accuracy obtained after training the representation model using different methods. For the methods in this table, the details are as follows:

(a) "Baseline" refers to training conducted using only real data, without any synthetic data.

(b) "Naive G2R" indicates that neither the reweighting nor the weak augmentation strategy is used; instead, synthetic data is directly mixed with real data to train the representation model.

(c) "G2R" is our default setting, incorporating the aforementioned two techniques.

(d) "PSL-10K + G2R" indicates that PSL is first used for 10K steps, and for the remaining training budget, the self-supervised learning method is employed. Similarly, "PSL-25K+G2R" conveys the similar meaning.

(e) All training costs are relative to the baseline.

From the results in Table 8 , it is evident that our default setting of G2R achieves better performance compared to the baseline and direct data mixing (Naive G2R). When PSL is first employed for training to a certain stage followed by self-supervised learning (see the last two rows of the table), a further performance improvement can be obtained. Additionally, we can achieve the same performance as the baseline with only 60% of the training cost (PSL-10K + G2R).

Table 8: Detailed results for efficient G2R training using pseudo-supervised learning.

| Method | Training Cost | Acc(%) |
|---|---|---|
| Baseline | 100% | $89.5 \pm 0.1$ |
| Naive G2R | 100% | $89.3 \pm 0.0$ |
| G2R | 100% | $90.9 \pm 0.0$ |
| PSL-10K + G2R | 60% | $89.5 \pm 0.0$ |
| PSL-10K + G2R | 100% | $91.8 \pm 0.0$ |
| PSL-25K + G2R | 100% | $91.3 \pm 0.0$ |

## C  EXPERIMENTAL DETAILS

### C.1  EXPERIMENTS ON R2G

**Acquisition of representation models with different capabilities.**  We train a representation model using the SimCLR method and selected models from $5\%$, $50\%$, and $100\%$ of the training stage as the weak, moderate, and strong versions of the representation model, respectively. Starting with these models, we conduct the R2G experiments.

For the SimCLR method used here, detailed configurations can be found in Table 9 . We use ResNet-50 as our backbone. For the lower-resolution images of CIFAR-10 and CIFAR-100, we replace the first convolutional kernel with a $3 \times 3$ convolutional kernel and removed the max pooling layer. All batch sizes represented the global batch size across all GPUs, and the learning rate is the effective value considering factors such as batch size and gradient accumulation steps. The reason for using different batch sizes for various datasets is that we allocated different amounts of computational resources, rather than for any more intricate reasons.

In the following experiments, unless stated otherwise, we use the backbone configuration of the representation models described here, along with the data augmentation settings detailed in the next subsection (see Table 12 ).

**Generative models.**  We employ ADM as the generative model for CIFAR-10 and CIFAR-100. The detailed configurations of ADM can be found in Table 10 .

For Tiny-ImageNet and ImageNet-1K, we opt for LDM due to the larger scale and higher resolution of these datasets. LDM, which carries out the diffusion process in a lower-dimensional latent space, is more computationally efficient than pixel-level diffusion models such as ADM. For the encoder and decoder of LDM, we use the pre-trained VQGAN [3], which is trained on the large-scale OpenImages dataset. The detailed configurations of LDM can be found in Table 11 .

For the results reported in Table 1 , we use the sampling method of DDIM (Song et al., 2020) to synthesize images, specifically 50 sampling steps, which significantly accelerates the image generation process.

In other experiments, unless stated otherwise, we employ the generative models and its configurations as described here.

---

[3]https://github.com/CompVis/taming-transformers#overview-of-pretrained-models

Table 9: Configurations of SimCLR for obtaining representation models with different capabilities.

| config | value | | |
|---|---|---|---|
| | CF-10/100 | T-IN | IN-1K |
| backbone | ResNet-50 | | |
| projection hidden dimension | 2048 | 4096 | 4096 |
| projection output dimension | 128 | 512 | 512 |
| temperature | 0.2 | | |
| optimizer | LARS | | |
| weight decay | 1e-4 | 1e-4 | 1e-6 |
| batch size | 512 | 1024 | 4096 |
| learning rate | 0.4 | 1.2 | 4.8 |
| randresizedcrop scale | [0.08,1.0] | | |
| color jitter - probability | 0.8 | | |
| color jitter - brightness | 0.8 | | |
| color jitter - saturation | 0.8 | | |
| color jitter - hue | 0.2 | | |
| gray scale | 0.2 | | |
| gaussian blur probability | 0.5 | | |
| horizontal flip probability | 0.5 | | |

Table 10: The configurations of the ADM.

| config | value |
|---|---|
| diffusion steps | 1000 |
| channels | 128 |
| residual blocks | 3 |
| channel multiplications | 1,2,2,2 |
| attention resolutions | 16,8 |
| head channels | 64 |
| learnable sigma | ✓ |
| noise scheduler | linear |
| BigGAN up/downsample | ✓ |
| EMA rate | 0.999 |
| dropout | 0.1 |
| loss function | $l_{simple}$ |

Table 11: The configurations of the LDM.

| config | value |
|---|---|
| latent vector shape | [4,32,32] |
| diffusion steps | 1000 |
| channels | 256 |
| residual block | 2 |
| attention resolutions | 32,16,8 |
| channel multiplications | 1,2,4 |
| head channels | 32 |

**Evaluation**    For the representation models, we evaluate them using the linear probing accuracy. We assess the generative models using FID and IS. We sample 50K images from the generative models, using the entire training set as the reference batch for CIFAR-10, CIFAR-100, and Tiny-ImageNet, with respective sizes of 50K, 50K, and 100K. For ImageNet-1K, we use the reference batch pre-computed and provided by ADM [4]. DDIM with 50 sampling steps is used for metrics evaluation.

## C.2   EXPERIMENTS ON G2R

**Synthetic data generation.**    We employ the DDIM sampling method, configuring the sampling steps to 50 to facilitate the rapid development of a slightly large synthetic dataset. During image generation, we uniformly apply the representation vectors from the reference training dataset. The synthetic dataset is configured to consist of 100K images. For instance, we iterate through the CIFAR-10 training dataset twice to facilitate the construction of the synthetic dataset.

**Reweighting and weak data augmentation.**    The ratio of real data to synthetic data, as well as the data augmentation strategies, can impact the performance of self-supervised learning methods. An appropriate ratio $\beta$ and weaker augmentation methods are beneficial for model learning (Wang et al., 2024b). However, a more refined theory is yet to be developed to analyze the relationship between these parameters and learning performance, and to guide their selection. Here, empirically, we selected the parameters shown in Table 12 for our experiments.

Table 12: Configurations of the reweighting ratio and data augmentation.

| config | value |
|---|---|
| reweighting ratio (CF-10/CF-100/T-IN) | 2 |
| reweighting ratio (IN-1K) | 1 |
| randresizedcrop scale | [0.2,1.0] |
| color jitter - probability | 0.4 |
| color jitter - brightness | 0.4 |
| color jitter - saturation | 0.4 |
| color jitter - hue | 0.1 |
| gray scale | 0.2 |
| gaussian blur probability | 0.5 |
| horizontal flip probability | 0.5 |

## C.3   EXPERIMENTS ON CORE

We adopt the default settings from the previous G2R and R2G configurations to conduct experiments on CORE across different datasets. As shown in Figure 5 and Table 6, under this configuration, we observe that both the generative model and the representation model experience co-evolution, with performance gradually improving as the rounds increase. Intuitively, since we do not aim for state-of-the-art performance but instead seek to explore the existence and effectiveness of co-evolution learning in this research, our hyperparameter configuration can be further improved through additional parameter searches.

## C.4   ABLATION STUDY

**Ablation experiments on different representational models.**    We conduct ablation experiments on four self-supervised learning methods and also experiment with supervised learning method using the cross-entropy loss function. For the baseline implementations of various self-supervised learning methods, we refer to the configurations from the open-source project solo-learn (Da Costa et al., 2022). Notably, BYOL uses asymmetric data augmentation in the baseline version, so in G2R, we use the corresponding asymmetric weak data augmentation. All self-supervised learning methods use ResNet-50 as the backbone, but the projector on top of it varies depending on the method.

---

[4]https://github.com/openai/guided-diffusion/tree/main/evaluations

**Ablation experiments on different generative models.** Three mainstream generative models (ADM, LDM, and DiT) are employed in the ablation experiments for generative models. These models represent pixel-level, latent-space-level, and transformer-based diffusion models, respectively. For ADM and LDM, we adopt the configurations described in Table 10 and Table 11 . For DiT, the model we use is DiT-L/2 with the pre-trained VAE tokenizer[5].

## C.5 EXPERIMENTS ON DATA-SCARCE SCENARIOS AND SELF-CONSUMING LOOP

**Configurations for the self-consuming loop.** The synthetic augmentation loop is the basic setup of our self-consuming loop, which requires that the data for the next round of generative model training comes from both the reference training dataset and the samples generated by the trained generative model in the previous round. Hence, for the baseline, we sample 50K images from the generative model of the previous round and combine them with 50K images from the reference training dataset, forming a new training set of 100K images. For CORE on the self-consuming loop, the construction of the training dataset for the generative model is consistent with the baseline. Additionally, we use the 100K sampled images to train the representation model via G2R.

**Configurations for the data-scarce scenarios.** We first randomly sample 1K images from each class in a class-balanced manner from the original CIFAR-10 training set of 50K images, forming a data-scarce dataset with a total of 10K training images, while keeping the test dataset the same as CIFAR-10. Consequently, only 10K images are used for the training of the generative model in R2G. We continue to sample 100K images from the generative model for training the representation model. Simultaneously, we adjust the reweighting ratio, increasing it to 10 to better accommodate the current data mixture, where real data is significantly less than synthetic data.

---

[5]https://huggingface.co/stabilityai/sd-vae-ft-ema

## D    SAMPLES FOR CORE

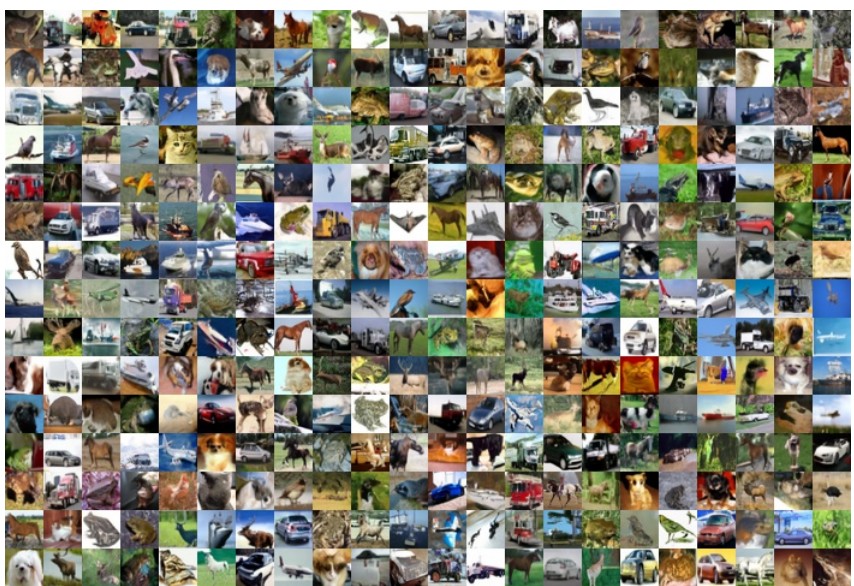

(a) Baseline (i.e., w/o CORE) (FID = 5.34)

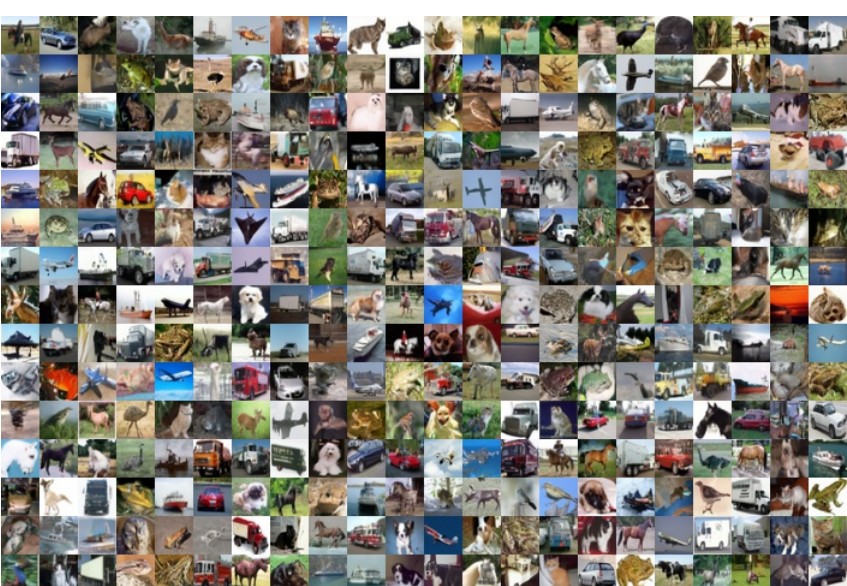

(b) The generative model after R2G (i.e., w/ CORE), corresponds to the "strong level" in Table 1 . (FID = 3.41)

Figure 9: Uncurated samples from CIFAR-10 32×32.

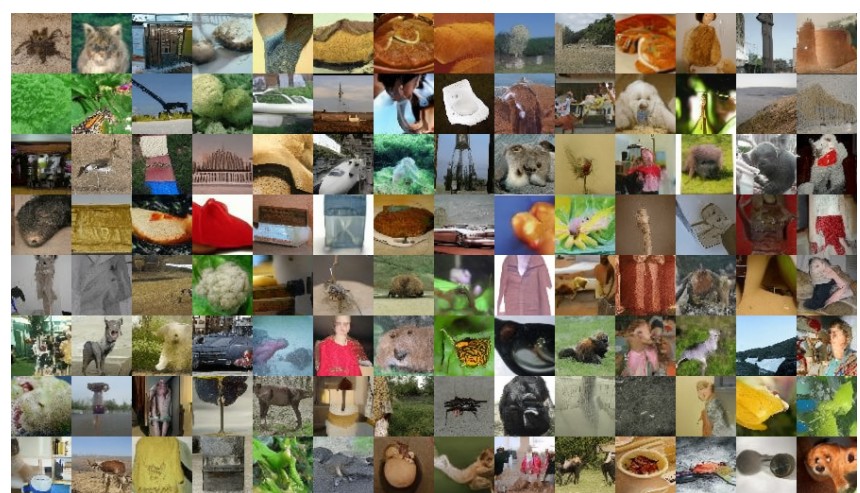

(a) Baseline (i.e., w/o CORE) (FID = 17.99)

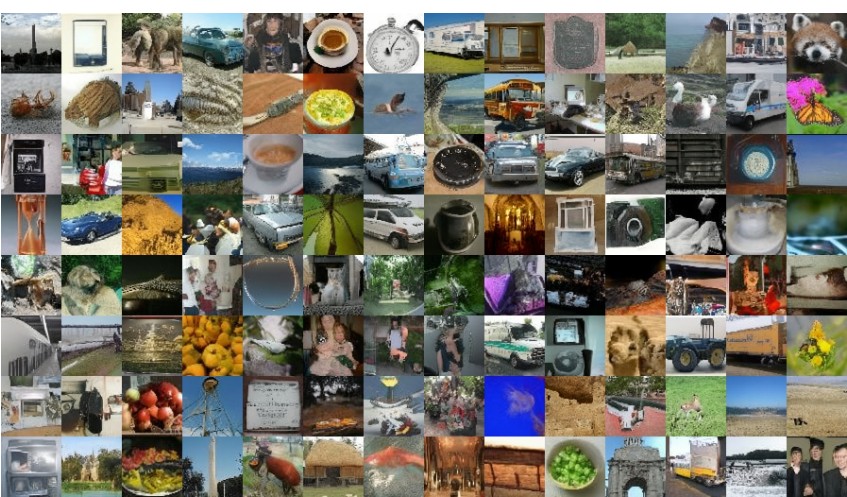

(b) The generative model after R2G (i.e., w/ CORE), corresponds to the "moderate level" in Table 1 . (FID = 12.29)

Figure 10: Uncurated samples from Tiny-ImageNet 64×64.

