# OpenReview forum: "Co-Evolution Learning"
_ICLR.cc/2025/Conference — Submitted to ICLR 2025_

### Official Review · Reviewer_pTXW · 2024-11-02

**Soundness:** 2
**Presentation:** 3
**Contribution:** 3
**Rating:** 5
**Confidence:** 4

**Summary:**

This paper tackles a key challenge in advancing generative and representation models: the dependence on high-quality, diverse data for training. To address these limitations, the authors introduce a co-evolution framework that enables generative and representation models to improve each other. Both representation and generation models progressively strengthen their performance by iterating through this mutual enhancement process.

**Strengths:**

1. The idea of co-evolution is interesting. It combines the two tasks in a unified framework and tries to help their corresponding model to improve each other in the mutual enhancement process.
2. The paper is well-organized, starting with a clear introduction of the current limitations and a detailed breakdown of the design of the proposed framework.

**Weaknesses:**

1. The use of a milder data augmentation strategy may have a limited impact on enhancing dataset diversity. Additionally, there is no ablation study to verify the effectiveness of this approach, even in Table 8, leaving its actual contribution to performance unclear.
2. An interesting observation in Table 2 is that using a weak generation model leads to a decline in the performance of the trained representation model. However, there is no analysis provided on this phenomenon or its potential risks, which would be valuable for understanding the limitations and stability of the proposed framework.
3. In the experiments across different datasets in Section 4.3, the generation model implementations vary, yet there is no clear explanations provided for these choices.
4. In the co-evolution experiments, it is unclear whether the generation model is trained from scratch or utilizes pre-trained generative capabilities. This lack of clarification makes it difficult to discern the true source of the observed training benefits.

**Questions:**

Please refer to the weaknesses.

---

### Official Review · Reviewer_GKML · 2024-11-02

**Soundness:** 2
**Presentation:** 2
**Contribution:** 2
**Rating:** 3
**Confidence:** 3

**Summary:**

This paper proposes a co-learning framework called CORE to jointly learn the representation and generative models. Specifically, it has two components, R2G framework which uses pretrained representation vision encoder to project data into latent space z, and learn a generative models by maximizing the log-likelihood conditioned on the z. The second component is G2R, which can sample diverse data points and can be used to learn a better latent representation. Experiments show that co-evolving these two components can facilitate the task performance for representation/generative tasks.

**Strengths:**

-- The proposed method empirically found that co-training can boost the performance of generative models training efficiency by 30%

-- The proposed Co-evolution of Representation modelsand Generative models (CORE) frame work is novel and interesting

**Weaknesses:**

--The paper is a bit hard to follow, for example, it is not clear what the main contribution of this framework after reading the introduction

--Experiments only conducted on small-scale dataset, CIFAR10/100 etc, where both SoTA generative models or representation learning methods already mastered and hard to tell if the performance come from parameter tuning or joint learning.

**Questions:**

-- How practical it is to implement this framework as the learning is iterative instead of end-to-end?

---

### Official Review · Reviewer_HzNh · 2024-11-03

**Soundness:** 3
**Presentation:** 3
**Contribution:** 1
**Rating:** 3
**Confidence:** 4

**Summary:**

In this work, the authors propose to learn simultaneously a representation model and a generative model following a mutual feedback loop. One path (R2G) uses the embeddings provided by the representation model to guide the learning of the generative model. The other path (G2R) leverages the generated images as augmented data to train the representation model. The combination of both is referred to as co-evolution (CORE). The experiments show that this setting improves the performance in both generative and representation models.

**Strengths:**

- The proposed approach is easy to understand, and provides moderate performance improvement.
- The paper is well structured and presented.
- Some experiments provide some useful insights.

**Weaknesses:**

- In my opinion the novelty is very limited. R2G is equivalent to an autoencoder with a pretrained and fixed encoder, and G2R is equivalent to an autoencoder with reconstruction loss in the latent space, i.e. $l_{rec}\left(\hat{z},z\right)$ with $z=f_1\left(x\right)$ and $\hat{z}=f\left(g\left(z\right)\right)$, where the first encoder and the decoder are pretrained and fixed. These settings and their combination (i.e. CORE) have been extensively used in the context of autoencoders and image-to-image translation models (and cross-modal translation models). [A-D] are some early examples that come to mind with similar setting. The main difference is the use of more modern generative models (diffusion), but that is not novel in my view.

[A] Unsupervised cross-domain image generation, ICLR 2017
[B] MUNIT: Multimodal Unsupervised Image-to-Image Translation, ECCV 2018
[C] Perceptual Generative Autoencoders, ICML 2020
[D] Mix and match networks: encoder-decoder alignment for zero-pair image translation, CVPR 2018

**Questions:**

Please address my concern about the novelty, and justify why the proposed model is significantly different from autoencoders with latent reconstruction loss.

---

### Meta-Review · Area_Chair_kf94 · 2024-12-25

**Metareview:**

The paper introduces a co-evolution framework (CORE) that jointly trains generative and representation models to enhance each other iteratively. The framework leverages semantic embeddings from representation models to improve the semantic consistency of generated data and utilizes diverse generated data to enrich representations. The reviewers question the paper in its novelty and experiment scale. The authors do not provide rebuttals to address these problems, leading to a decision to reject this paper.

**Additional Comments On Reviewer Discussion:**

No rebuttal is provided.

---

### Decision · Program_Chairs · 2025-01-22

Reject